# Effect of Phase-Change Materials on Laboratory-Made Insoles: Analysis of Environmental Conditions

**DOI:** 10.3390/ma15196967

**Published:** 2022-10-07

**Authors:** Elena Arce, Rosa Devesa-Rey, Andrés Suárez-García, David González-Peña, Manuel García-Fuente

**Affiliations:** 1Department of Industrial Engineering, Polytechnic School of Engineering of Ferrol, University of A Coruña, 15403 A Coruña, Spain; 2Defense University Center, Naval Academy, University of Vigo, Plaza de España 2, 36920 Marín, Spain; 3Research Group Solar and Wind Feasibility Technologies (SWIFT), Electromechanical Engineering Department, University of Burgos, 09006 Burgos, Spain

**Keywords:** PCM, insole, epoxy resin, paraffin, insole thermal properties, Box–Behnken design

## Abstract

Thermal comfort is essential when wearing a postural-corrective garment. Discomfort of any kind may deter regular use and prolong user recovery time. The objective of this work is therefore to optimize a new compound that can alter the temperature of orthopedic insoles, thereby improving the thermal comfort for the user. Its novelty is a resin composite that contains a thermoregulatory Phase-Change Material (PCM). An experimental design was used to optimize the proportions of PCM, epoxy resin, and thickener in the composite and its effects. A Box–Behnken factor design was applied to each compound to establish the optimal proportions of all three substances. The dependent variables were the Shore A and D hardness tests and thermogravimetric heat-exchange measurements. As was foreseeable, the influence of the PCM on the thermal absorption levels of the compound was quantifiable and could be determined from the results of the factor design. Likewise, compound hardness was determined by resin type and resin-PCM interactions, so the quantity of PCM also had some influence on the mechanical properties of the composite. Both the durability and the flexibility of the final product complied with current standards for orthopedic insoles.

## 1. Introduction

Thermal insulation in footwear is one of the most relevant properties for user comfort that should be considered during the design phase. High in-shoe temperatures heighten the risk of fungal and bacterial infections and can cause discomfort to users, who may refuse to wear such footwear. The use of insoles generally augments in-shoe temperature, which can exacerbate the above-mentioned problems.

It has been shown in several studies that temperatures between 27–33 °C are comfortable for users, while temperatures above 35 °C are likely to produce discomfort [1]. Regulation of in-shoe temperatures may be achieved with a Phase-Change Material (PCM) that can be specifically designed to undergo a change of phase whenever the temperature drops below or rises above 35 °C. Fluctuating temperatures can therefore change the state of a PCM, so that the heat that builds up within the shoe is never excessive.

A PCM is a substance within which energy is exchanged through an isothermal process that implies a physical change in state [2]. One property of a PCM is that energy is retained within the material in the form of latent heat during the melting cycle, which can be used to control temperatures within a certain range [3]. As the PCM solidifies, it releases retained energy, demonstrating its capacity to retain excess heat in liquid form and to discharge it into the ambient atmosphere as it solidifies. This latent heat retention capacity has increased interest in the development of PCM of different sorts, due to the properties that they provide. Therefore, a PCM may be used in new composite materials designed to achieve thermal stability. Technological advances, new designs, and structural needs can at times imply a need for new materials in response to new demands [4].

In the textile sector, there are scarce few references to footwear, as previous studies have mainly been focused on temperature regulation systems for clothing [5,6,7]. Foot thermal comfort is strongly dependent on temperature and humidity [8]. The analysis of different experimental conditions that might affect in-shoe comfort is important to prevent fungal and bacterial foot infections [9]. Other parameters such as mechanical properties, the intensity of an activity, ambient conditions, and sweating are also important parameters for determining thermal regulation in footwear [10]. Some attempts to increase comfort have included the manufacture of 3D thermoplastic polyurethane insoles, which showed good humidity control around both sole and heel, though no temperature changes were noted [11]. Ferraz et al. [12] also tested temperature dissipation with Peltier-based cooling system modules built into the shoe structure. The contribution of our study is to thermal regulation in footwear through the addition of a PCM that can be used in traditional manufacturing technologies. The PCM microcapsules are encapsulated within an epoxy resin and contribute to heat regulation without affecting the mechanical properties of the insole. In an earlier study [13], PCM microcapsules tested in cold-weather clothing were shown to reduce thermal conductivity by approximately 13% while significantly improving thermal insulation, thereby improving temperature regulation without employing heavy, multilayered clothing systems.

In this study, the design matrix of the composite PCM for regulating the temperature of orthopedic insoles was a mixture of an encapsulator, a thickener, and a PCM. An epoxy resin was employed as the encapsulator: a thermosetting polymer with a high level of crosslinking that prevents solubility [14], a typical epoxy resin that was produced by the condensation of bisphenol A and epichlorohydrin, with hydroxyl pendant groups and terminal epoxy groups [15]. During its formation, a hardener was added that converts the epoxy resin into a highly crosslinked insoluble polymer, with a high melting point, and with characteristics quite unlike its previous form [16]. As opposed to other chemical bonding reactions, this three-dimensional network forms very strong bonds that compact even further whenever the temperature increases, forming a stronger polymeric structure. Recent studies have underlined the importance of developing composite materials to improve the properties of epoxy glues used in footwear. There is now a patented composite carbon and resin-layered material, to prevent deterioration due to environmental factors such as temperature changes, humidity, and solar radiation [17]. The results showed good wear resistance due to the high mechanical strength of the carbon layers, although no mention of temperature control was reported. Nevertheless, good thermal stability was observed in epoxy resins modified with Epoxidized Hydroxyl-Terminated PolyButadiene (EHTPB). Improved mechanical performance was also observed [18], although the resin was neither applied nor tested in footwear. 

The manufacture of personalized orthopedic insoles forms part of the podiatry sector. Feet differ as do podiatric pathologies and problems. Corrective insoles are intended to support the foot arch and heel, thereby improving plantar stability and relieving foot supination. They are made of different materials (e.g., thermoplastics, polyethylene foams, leather, cork, composite carbon fibers, ethyl-vinyl acetate (EVAS), gel) [19]. Thermoplastics combined with polyethylene foams are the most widely used insole materials. The vacuum formed thermoplastic material (usually polypropylene, resin, or carbon fiber), is placed under the foot in the arch and heel area for support and gait correction. Foam is placed in the forefoot area and may cover the entire insole. Insoles of this type are extremely thin and can easily be adapted to any type of footwear. A careful diagnosis of foot ailments, gait posture, and related foot pain is necessary for proper insole development. 

The main objective of this work is to incorporate PCM in orthopedic insoles without modifying any physical characteristics of the insoles, so as to improve in-shoe thermal properties. It was therefore necessary to search for a thermo-regulatory composite with the same hardness as standard insole materials that a podiatrist might use for the development of orthopedic insoles. Three main substances form the composite: an encapsulator in the form of an epoxy resin, a thickener, and the PCM formulation and testing of this new material was necessary as a first step towards the manufacture of the composite material for temperature regulation within orthopedic insoles. When the in-shoe temperature increases, the PCM absorbs the heat that is used as latent energy in the solid-to-liquid phase change. Conversely, whenever there is a decrease in temperature, the PCM will release energy, causing a change of state from liquid to solid. In this way, the thermal regulation within the shoe is achieved through either the absorption of excess heat or the release of accumulated heat. The characterization of the PCM and the main results are presented and discussed in the following sections.

The novelty of this study is linked to the development of a moldable thermoregulatory epoxy resin for the development of orthopedic insoles that support both heel and foot arch. A major disadvantage of orthopedic insoles, which deters their use among some patients, is the significantly higher in-shoe temperature, affecting other relevant variables such as perspiration [20] and fungal infection. In Figure 1 (left), a schematic representation of thermal energy flows is presented. The foot cannot effectively exchange heat with the outside environment via radiation or convection because of the thermal insulation of the orthopedic insole. As a result, the only method of regulating body temperature is sweating [10]. The PCM helps to mitigate this problem through the absorption or the dissipation of heat whenever a phase change takes place (Figure 1, right).

An innovation in the footwear industry is therefore presented in this study, as the use of a composite material made of PCM, epoxy resins, and thickeners can be used to regulate the in-shoe temperature, regardless of physical activity and ambient temperature. The final product, in addition to ensuring thermal comfort within a shoe, must be sufficiently durable to withstand mechanical stress.

## 2. Materials and Methods

In this study, two different epoxy resins were tested: Resoltech 1600 and Resoltech 1050 (from Castro Composites, Pontevedra, Spain). Resoltech 1600 is an epoxy resin with flexible properties, air temperature curing, and high impact resistance, whereas Resoltech 1050 presents exceptional wetting properties. A mixture of Resoltech 1600 and 1050 resins produces laminates that present high flexibility and high impact resistance. In addition, the flexibility of the mixture can be adjusted by varying the percentage of Resoltech 1050 resin in the sample. Frequently employed in mixtures containing epoxy, polyester, and vinylester resins, the third element of the mixture is a thickener to improve the mechanical properties. Thickeners are related to the use of novel molding materials for producing coating materials, adhesives, casting resins, flame retardants, agents with thixotropic effect, and/or reinforcing agents [21]. Added in low concentrations, typically 0.5–5%, their main purpose is to provide higher viscosity and thixotropy. Having added the thickener to the mixture, the PCM (paraffin wax) is more easily encapsulated within the epoxy resin, without affecting the resin gelling time. In addition, the thickener contributes to the production of a totally homogeneous mixture, as it shows an anti-settling behavior and high bulk density. The thickener used in this study, based on organophilic phyllosilicates, was Garamite 1958 (Thickener 004) from Castro Composites, Pontevedra, Spain. Rubitherm RT 35HC paraffin (Rubitherm, Berlin, Germany), an organic phase-change material, was used as the PCM. Paraffin has a high energy retention capacity. The solid-to-liquid melting process takes place at an almost constant temperature. Rubitherm RT 35HC paraffin wax, a chemically inert wax, can either retain or dissipate 240 kJ/kg [22], without undergoing any chemical reactions. In addition, its long lifespan and phase-change temperature of 35 °C [22] make it suitable for the application under study.

The multi-component sample solution was developed in six steps: (1) Mixing of resin and hardener; (2) Weighing and addition of PCM; (3) Weighing and addition of thickener; (4) Mixing (stirring) components until they formed an homogeneous matrix; (5) Pouring the matrix into the molds; and (6) Curing of the matrix. A flowchart of the three-stage process followed to develop the samples in this research is shown in Figure 2: (1) Pre-characterization; (2) Optimization of composition; and (3) Verification. In Stage 1, in-shoe temperatures during different types of physical activity practiced at different intensities were determined using sensors installed within the shoe. At the same stage, the shore D hardness of an orthopedic insole used as a reference was measured. The (resin-hardener) mixture composition had to be determined for the preparation of the samples. After the hardener was added, the epoxy resin was converted into an irreversible polymer, with characteristics quite unlike its previous viscous liquid state. The composite manufacturer specifies the proportions of resin and binder that are required, depending on specific mechanical and thermal properties. In this initial study, a thermogravimetric analysis determined the thermal properties of the resin with different hardener ratios. In Stage 2 (Composition optimization), a Box–Behnken factor matrix was used to determine the effects of each factor on the response variable that was measured in terms of energy exchange. Having obtained the results of the thermal analysis, the optimal sample could then be determined from a statistical analysis of the variables. In Stage 3, various tests (long-duration analysis, Scanning Electron Microscopy (SEM, JEOL, Tokio, Japan), and sweat analysis) on the optimal sample selected in Stage 2 were applied, in order to check its suitability.

### 2.1. Preliminary Study

Before embarking on the design and the preparation of the orthopedic insoles, a preliminary characterization of some commercial insoles, including their hardness and in-shoe temperatures, was conducted. Temperatures within the shoe were measured with two sensors. The first one was employed to determine the room temperature and the other was positioned within the shoe. A correlation of both sensor readings determined the influence of the external ambient conditions on the internal temperature within the shoe. *iButton* DS1922L-F5, Whitewater, WI, USA (range—40, 85 ± 0.5 °C) sensors that record temperature readings at 10-min intervals were employed. Finally, hardness was evaluated on conventional orthopedic insoles using laboratory-made probes. For this purpose, three orthopedic insoles were manufactured to evaluate their properties without the PCM as an additive. The insoles were manufactured with a 100% resin and hardness was measured through the Shore D test (PCE-DX-DS, PCE Iberica Albacete, Spain) at five points on the insoles (Figure 3). It should be noted that these are orthopedic insoles, so the required durability and flexibility of the material will depend on the level of material hardness. The greater the hardness, the greater the corrective or restraining effect. The hardness of the material (resin) is chosen according to the pathology and the daily activity of the person, i.e., a harder resin for more intense physical activity. An insole designed for containment and intensive daily use was used as a reference (Figure 3). As the insole was made of a hard material, it was measured with the Shore D hardness scale. In addition, another four samples of ten grams were made with a high PCM percentage, to check the PCM concentration and saturation levels.

### 2.2. Study of the Optimal Insole Composite Material and Its Properties

In this study, an incomplete 3^3^ factorial design was conducted, to elucidate the optimal combination of the three substances—epoxy resin, thickener, and PCM—that constitute the PCM-dosed insoles, which should regulate in-shoe temperature without affecting insole hardness. The relationship between coded and uncoded variables was established by linear equations deduced from their respective variation limits, according to Equation (1) [23]:(1)$$x_{i}=\left(\frac{Z_{i}-Z_{i}^{0}}{\Delta Z_{j}}\right)\beta_{d}$$
where $\Delta Z_{i}$ is the distance between the real value in the central point and the real value in the superior or inferior level of a variable; $\beta_{d}$ is the major coded limit value in the matrix for each variable; and $Z^{0}$ is the real value at the central point. Coded variables were then assigned values of −1, 0, and +1, corresponding to the lowest, central, and maximum limits of variation for each variable. The magnitude of each variable therefore had no influence on the response surface obtained from the coded variables through which the factors were combined on a dimensionless scale.

Ten-gram samples of orthopedic insoles were manufactured in the laboratory to evaluate the influence of the three variables proposed in this study. An incomplete factorial design was applied to establish the optimal formulation of the manufactured orthopedic insoles [23,24]. The independent variables and the limits of variation were defined as R (percentage of resins 1600 and 1050 employed), T (mass of thickener), and PCM (mass of phase change material). Standardized (coded) dimensionless independent variables were used, with variation limits (−1, 1), defined as $x_{1}$ (coded dose of resin), $x_{2}$ (coded dose of thickener), and $x_{3}$ (coded concentration of PCM employed). The dependent variables consisted of the hardness measured for each orthopedic insole that was analyzed: $y_{1}$ (Shore D hardness), $y_{2}$ (Shore A hardness), $y_{3}$ (absorbed heat after thermogravimetry) (Table 1).

Data were analyzed using the STATGRAPHICS statistical program (Version 5.1 Statistica for Windows; StatSoft, Inc., Tulsa, OK, USA), and a quadratic function was obtained for all the dependent variables in the study (Equation (2)).
(2)$$y=\beta_{0}+\beta_{1}x_{1}+\beta_{2}x_{2}+\beta_{3}x_{3}+\beta_{12}x_{1}x_{2}+\beta_{13}x_{1}x_{3}+\beta_{23}x_{2}x_{3}+\beta_{11}x_{1}{}^{2}+\beta_{22}x_{2}{}^{2}+\beta_{33}x_{3}{}^{2}$$
where y is the dependent variable, β denotes the regression coefficients (calculated from experimental data by multiple regressions using the least-squares method), and x denotes the independent variables corresponding to $x_{1}$ (resin dose), $x_{2}$ (thickener dose), and $x_{3}$ PCM concentration).

Table 2 shows the set of experimental conditions assayed for the independent variables (expressed in terms of coded variables). Table 2 also lists the dependent variables, $y_{1}$, $y_{2}$, and $y_{3}$, which correspond to Shore D and Shore A hardness measurements, and adsorbed heat during use, respectively. In all, 15 experiments were performed to evaluate the influence of each independent variable on variations of both hardness and heat exchange. The absorbed heat variable (Table 2) was used to compare the thermal potential of the different samples under development. Experiments 13−15 were replicates of the central point of the design, employed as statistical control to evaluate the experimental error. Analysis of replicates of the central point (experiments 13−15) showed a low variability, with variation coefficients lower than 10% in all cases.

The next step was to determine the dependent variables. Shore D and Shore A tests were applied to determine the hardness of the samples for proper measurement throughout the solidification process of the PCM materials. Shore A tests are more suitable for softer, flexible materials, whereas Shore D tests are primarily designed for harder materials. Fifteen cycles of thermogravimetry (Labsys Evo, SETARAM Instrumentation determination, Labsys company, Redon, France) analysis over two hours determined any mass loss. The design of the cycle consisted of a total of three warm-up and cool-down periods: it started with a constant temperature at 20 °C for 300 s to reach a stable temperature; next, the heating and cooling ramps had a slope of 2 °C/min with a maximum temperature of 50 °C. The first cycle results were discarded due to thermal inertia presented by the crucible, which caused some distortion of the results. Measurements were taken in triplicate for better accuracy.

### 2.3. Verification Tests on the Optimal Sample

Having obtained the optimum values, the following verification tests were performed: (1) thermogravimetry study after subjecting the specimen to sweating conditions for 48 h with a synthetic solution adjusted to pH 4.5; (2) prolonged thermogravimetric testing of the optimized specimen over ten additional cycles and for 6 h; (3) SEM of the sample that had been pretreated with a thin carbon-sputtered layer to increase conductivity.

The objective of the experiments was to evaluate specimen behavior under prolonged real-life conditions. Additionally, Differential Scanning Calorimetry (DSC) of the sample was used to evaluate the amount of heat that the material had either released or absorbed. This energy exchange can occur when a certain sample is heated or cooled for a period of time. The temperatures of the sample and of the internal reference of the equipment were compared to obtain the value of the heat flow.

## 3. Results and Discussion

### 3.1. Previous Characterization of the Material

A previous characterization was performed, in order to establish the base parameters of the orthopedic insoles prior to use. Temperature measurements made with the help of the *iButton* sensors are presented in Figure 4. For simplicity, only day 3 results are presented, as no significant variations were observed during the period that was analyzed. The graph depicts a comparison between the temperatures within the room and within the shoe, showing the high levels of energy that the human body releases during physical activity. In the first part, it can be seen that the in-shoe temperature exceeded the ambient temperature by almost 10 °C, following some high intensity exercises of the user, when the phase-change temperature of the paraffin, 35 °C, was reached. It shows the influence of user activity on the temperature within the shoe.

Hardness tests were applied to the three shoe insoles manufactured specifically for this study (Figure 5). Three measurements were taken at each of the five different points in the central part of the insole (Figure 3) and results are shown in Table 1. Mean Shore D values were 55.7, with no significant variations between specimens and sampling points, and a variation coefficient under 10% in all cases. These results will be employed to compare with laboratory-made insoles where the effects of the PCM, epoxy resin, and thickener on the samples will be evaluated. Hardness results were consistent with other results found in the literature [25,26] so manufactured insoles could be employed for further analyses.

In addition, a preliminary study was performed by mixing different ratios of the two resins evaluated in this study (i.e., Resoltech 1600 and Resoltech 1050) and the commercial thickeners (i.e., Resoltech 1606 and Resoltech 1056) to evaluate the mechanical and thermal properties of these samples. The test ranges were selected according to the data sheet of the commercial materials. These mixtures gave values of Shore D hardness within the range of the commercial insoles: 49, 58, and 64. As a part of the previous thermal characterization, energy release samples were also analyzed in the laboratory-manufactured insoles by sampling curing at 60 °C for 16 h. Analyses performed for 100% resin samples showed a minimal energy exchange, in the range of 32–34 °C.

These results were fed into the Box–Behnken factor design, in order to elucidate the optimum concentrations of the variables that were analyzed. The following step was to test several combinations to achieve the optimal properties.

### 3.2. Optimization of the PCM Concentration in Orthopedic Insoles

The set of experimental conditions assayed (expressed as coded variables) and the experimental data obtained for variables y_1_ to y_3_ are shown in Table 3. The results showed different effects on hardness and heat exchange, with Shore A hardness values that ranged from 93.4 to 97.6; Shore D varied between 49 and 59.4, and heat exchange varied between 0.1893 and 0.6708 µV/mg. Therefore, Shore A hardness was the least sensitive parameter to optimization, showing variations of around 5% between experiments. The experimental results of the Shore D measurements varied slightly more at around 8%. On the contrary, heat exchange showed differences between experiments of up to 30%, which confirms the importance of parameter optimization in the development of thermal-comfort materials. It can also be inferred from the results that the samples with the highest PCM concentrations presented the highest heat exchange indices, although the correlations with hardness had to be analyzed with statistical tools.

The effects of the standardized independent variables on the dependent variables under study are shown in Figure 5. The largest bars indicate the most important variables for temperature regulation within the shoe. In the Pareto chart, Shore D hardness is shown to have a dependency on the resin concentration, which is the only parameter influencing hardness, whereas the other independent variables had no significant influence on this property. In turn, Shore A hardness (Figure 5) also showed some dependency on the resin concentration, although the combination of the epoxy resin and the PCM also showed a hardening effect. Finally, enthalpy showed a dependency that was exclusively based on the PCM concentration, with a positive effect, confirming that the PCM favored heat exchange. The above effects can also be visualized in the Principal Effects Graphs (Figure 6). Resin was the most influential variable for Shore D hardness, with higher resin concentrations associated with increased hardness. The maximum point on the curve indicates that the highest effect was reached for Shore D hardness. In contrast, the positive linear trend of the representation of Shore A hardness might suggest that an increased resin percentage could still lead to increase higher Shore A hardness values. Finally, the highest effect in terms of heat exchange was observed for the PCM dose, which showed a positive effect as PCM increased.

Based on the above results, predictive equations can be formulated with the Box–Behnken factor design, which can then be employed to determine the most influential variables, in terms of linear, interactive, and quadratic factors. The significance of each coefficient was determined by *p*-values for the variables y_1_ to y_3_, corresponding to the additions to the mixture of quantities of resin, thickener, and PCM, respectively. Using these coefficients, equations can be formulated, in order to determine the values of the dependent variables within the test ranges that are studied. Equations (3)–(5) may serve, respectively, as proxies for the formulation of orthopedic insoles with PCM to regulate temperature change, obtained with the optimum combination of resin, thickener, and the paraffin-wax PCM:*Shore D hardness (y_1_) = 96.0 + 1.25[R] − 1.1[R]^2^*(3)
*Shore D hardness (y_2_) = 52.2 + 4.65[R] + 1.55[R][PCM]*(4)
*Enthalpy (y_3_) = 0.4 + 0.15[PCM]^2^*(5)
where [R] is the quantity of resin employed and [PCM] is the dose of phase-change material needed for effective temperature control. The above equations were deduced, discarding any coefficients with an insignificance level at a confidence level of 95% (*p* > 0.05).

Having determined the most influential factor on hardness and heat exchange, the effects of increased concentrations of both resin and PCM could be considered, in order to infer the effect on the final insole design. An ascending response was modelled in order to predict the variation of the dependent variables at some distance from the experimental range under assay. Table 4 shows the path from the center of the current experimental region along which the estimated response rapidly varies with minimal variation of the experimental factors. Whether the goal was either to increase or to decrease hardness and enthalpy, the results revealed good locations for additional experiments. Six points were generated, increasing the content of resin at increments of 1.0%. Increased resin concentrations of up to 5% could therefore yield a Shore D hardness of 402.386, whereas Shore A hardness could reach 104.431 and heat exchange could be maximized up to −41.9638 µV/mg.

Finally, the combination of independent variables may be employed to determine the optimum conditions, which are expressed in Table 5, in terms of coded and uncoded values. The uncoded—real—values of the independent parameters should be employed as follows: 40% of resin, 0.48 g of thickener, and 2 g of PCM, in order to improve Shore D hardness; 40% of resin, 0.3 g of thickener, and 2 g of PCM to improve Shore A hardness; and 20% of resin, 0.3 g of thickener, and 1.9 g of PCM to increase heat exchange.

It is particularly interesting to examine both thermal analyses and endothermic enthalpies, as the findings may be employed to infer the results of experiments where energy is retained. Figure 7 shows the exo- and endothermic transformations over two reversible cycles of a ThermoGravimetric-Differential Scanning Calorimetry (TG-DSC) heat analysis. Glass transition and degradation occur at almost constant rates, the slowest of which comes before the solid state. Glass transition occurs after solidification and at the beginning of the melting process, while degradation occurs at the end of the melting process and at the beginning of solidification. Solidification is an exothermic process (above the red dotted line in Figure 7), while melting is an endothermic process (below the red dotted line in Figure 7). When a sample substance is subjected to thermal cycles, the effects of heat can produce changes in its properties. Variations in mass are analyzed with a thermogravimetric analyzer, a precise thermobalance for isothermal and dynamic tests. Energy changes (endothermic or exothermic) are analyzed with Differential Scanning Calorimetry (DSC). The DSC technique measures the heat flow within a sample which is then compared with a reference sample. The Labsys evo TG-DTA DCS (Labsys company, Redon, France) was used in this study. To do so, it was necessary to calibrate the equipment with reference substances. The dashed red line indicates the variations of sample mass during the development of the test; the green line represents the energy flow: a positive value suggests an exothermic process, i.e., energy is released; in contrast, negative values indicate that the sample is undergoing an endothermic or energy absorption process. Oven temperature is represented by a blue line and the first cycle was discarded due to thermal inertia.

### 3.3. Behavior of Samples under Real, Prolonged Conditions

According to the optimum values that are explained in Section 3.1, sample 12 was selected as the most suitable specimen corresponding to the optimized conditions for all three dependent variables. The weight of an orthopedic insole will vary, depending on foot widths and type of corrective treatment: an average weight of 50 g of resin per insole was estimated. Sample 12, the selected sample, contained 20% by weight of PCM. Considering the properties of paraffin, that weight implies a latent heat in the orthopedic sole of approximately 48 kJ/kg. These data are an estimation, and the final prototype still has to be manufactured and tested.

The sample was subjected both to different mechanical stresses and to thermal variations that affected its physical and chemical characteristics. Sweating tests were performed to simulate real conditions of the insoles. After 48 h of time in contact with the solution, the specimens were examined under the microscope and no damage was observed. The results of the thermogravimetric analysis showed that the enthalpy measured in the sample before the sweat test was 0.6211 µV/mg, whereas the sweat-test value after 48 h was 0.6022 µV/mg. A variation of approximately 3% between both measurements was observed.

Prolonged thermogravimetric analyses were also applied to the sample and no signs of damage were observed. Moreover, the sample showed the same amount of energy released during the second cycle as the original (0.5990 µV/mg). However, an increasing trend in the energy that was released should be noted after each of the cycles ended. From the second to the fifth cycle the value increased up to 0.7852 µV/mg to stabilize at 0.8298 µV/mg after the fifth cycle. It can be clearly seen how the enthalpy values stabilized as from the middle of the test, so the idea that the material lost properties over time could be ruled out. SEM analyses of the samples (Figure 8) presented homogeneous microscopic structures, with an absence of roughness and cavities, suggesting that the mixing, curing, and post-curing processes had been properly performed and that all the additives had been homogeneously combined. Figure 8 shows the sample surface with a scale of 300 µm on the left-hand side and with a scale of 500 µm on the right-hand side. In these two images, the uniform distribution of the additives can also be observed. The points of higher brightness correspond to the presence of materials with higher atomic numbers, while the darker areas have lower atomic numbers.

The results underline the potential use of PCM to control heat exchange within commercial insoles and the use of epoxy resins as a material to control heat exchange [27,28,29]. There are few references to PCM in the context of footwear [30,31] in which different materials are employed for heat control. However, it is a crucial factor, as an adverse thermal microenvironment within footwear results in foot-related thermal discomfort and hygiene problems [32]. A correlation was also found between fungal and bacterial growth on foot plantar skin and in-shoe temperature [33]. Similarly, penetration of PCM in the textile industry is still relatively limited, although some references refer to the achievement of temperature control between 20 and 28 °C [34] and other studies have researched their capacity to maintain flexibility and breathability, among others [35]. The potential application of PCM to footwear has shown promising advances in relation to thermal regulation and the stability of the synthesized materials and the results will contribute to research on new materials that may increase user comfort.

## 4. Conclusions

In this study, the use of Phase-Change Materials (PCM) to improve thermal comfort in insoles has been analyzed. For this purpose, tests have been performed on two epoxy resins—Resoltech 1600 and 1050—together with variable concentrations of Garamite, employed as thickener. The influence of the mixture on the final hardness of the samples and appropriate heat exchange between the insoles and the ambient atmosphere have all been tested. A verification of the synthesized materials has also been conducted, by means of prolonged assays to ensure the durability of the materials. The results have shown that the addition of PCM to the samples had no effect on hardness, with values ranging from 49 and 59.4 (Shore D) and 93.2 and 97.6 (Shore A), whereas heat exchange reached values up to 0.67 µV/mg. Based on the results, curing was key to the thermal and mechanical properties of the samples. The optimization test results have shown that heat exchange depended only on the PCM concentration, whereas hardness was also dependent on resin concentrations. An ascending response was modelled to show that by increasing resin concentrations up to 5%, hardness could increase over 400 (Shore D) or 100 (Shore A), whereas heat exchange could increase up to 41.9 µV/mg. Although adding more PCM to the samples resulted in higher enthalpy levels, it led to poorer mechanical properties. Finally, durability tests were performed by submitting the materials to ten cycles of use and the results showed that heat exchange had no effect on the material and no signs of structural damage were observed when examined with SEM.

Finally, a possible future line of research is the analysis of thermal comfort. As indicated above, this is an aspect of particular relevance in both the clothing and the footwear industries. The complexity of this process makes it necessary to develop new study methods. Although the analysis of heat flow is relevant, so too is wearer-perceived thermal comfort (e.g., in-shoe temperature, sweating). As may be expected, this complex step is necessary before a commercial product is produced. However, it is due to its complexity that it is still seen as a future line of work.

Likewise, thermal comfort was not analyzed in this study. Thus, as another future line of work, we propose the study of the thermal performance of the sole of the shoe with and without insoles.

## Figures and Tables

**Figure 1 materials-15-06967-f001:**
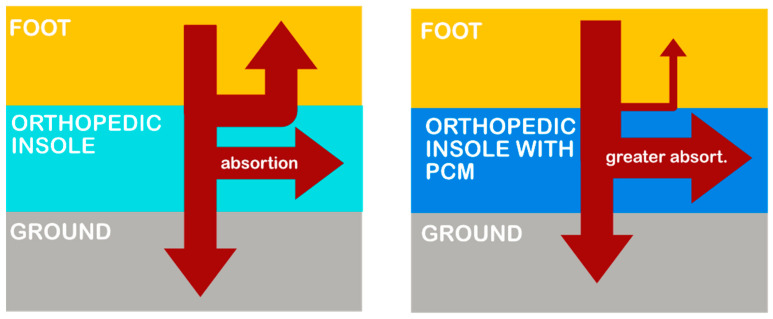
Heat transfer pathways between foot and ground: (**left**) traditional orthopedic insole; (**right**) proposed orthopedic insole with PCM.

**Figure 2 materials-15-06967-f002:**
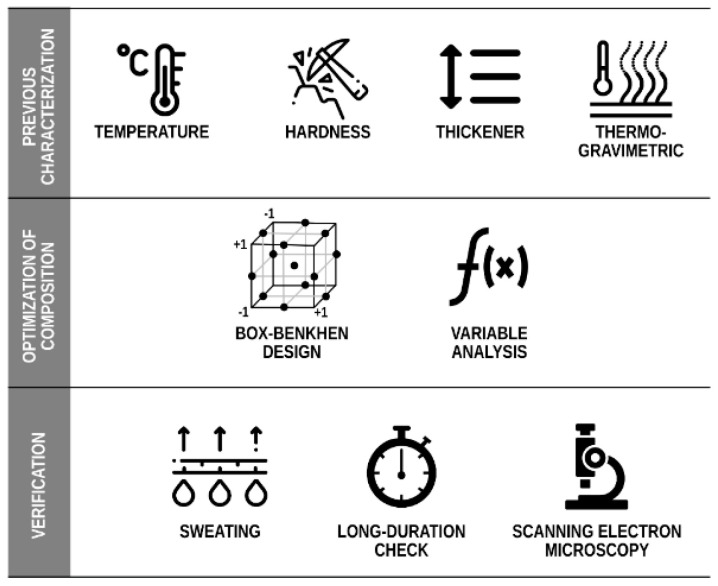
Procedure followed to formulate the new insoles.

**Figure 3 materials-15-06967-f003:**
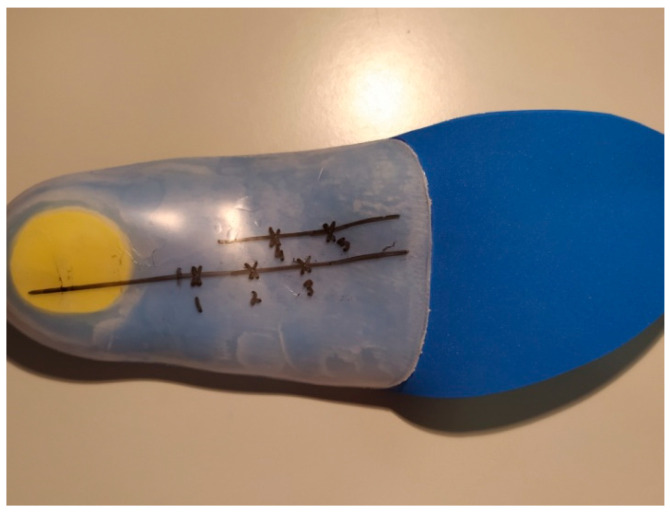
Orthopedic insoles manufactured for this study (crosses marked in black show the five Shore hardness measurement points).

**Figure 4 materials-15-06967-f004:**
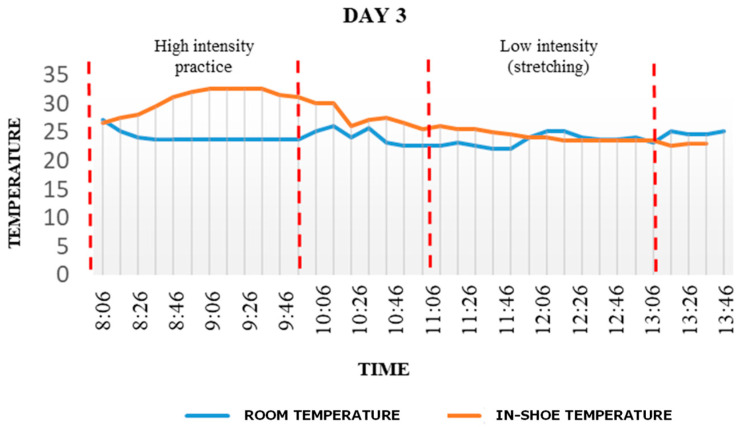
Temperature variations on Day 3 depending on the activity. Room temperature is shown with a blue line and in-shoe temperature with an orange line.

**Figure 5 materials-15-06967-f005:**
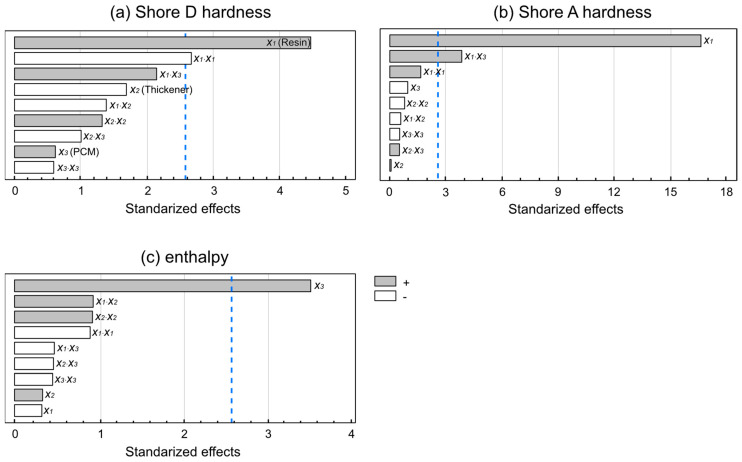
Pareto chart of standardized effects, showing the influence of the independent variables on the dependent variables. (**a**) Shore D hardness; (**b**) Shore A hardness; (**c**) Heat exchange. Values exceeding blue line are those significantly influencing the process.

**Figure 6 materials-15-06967-f006:**
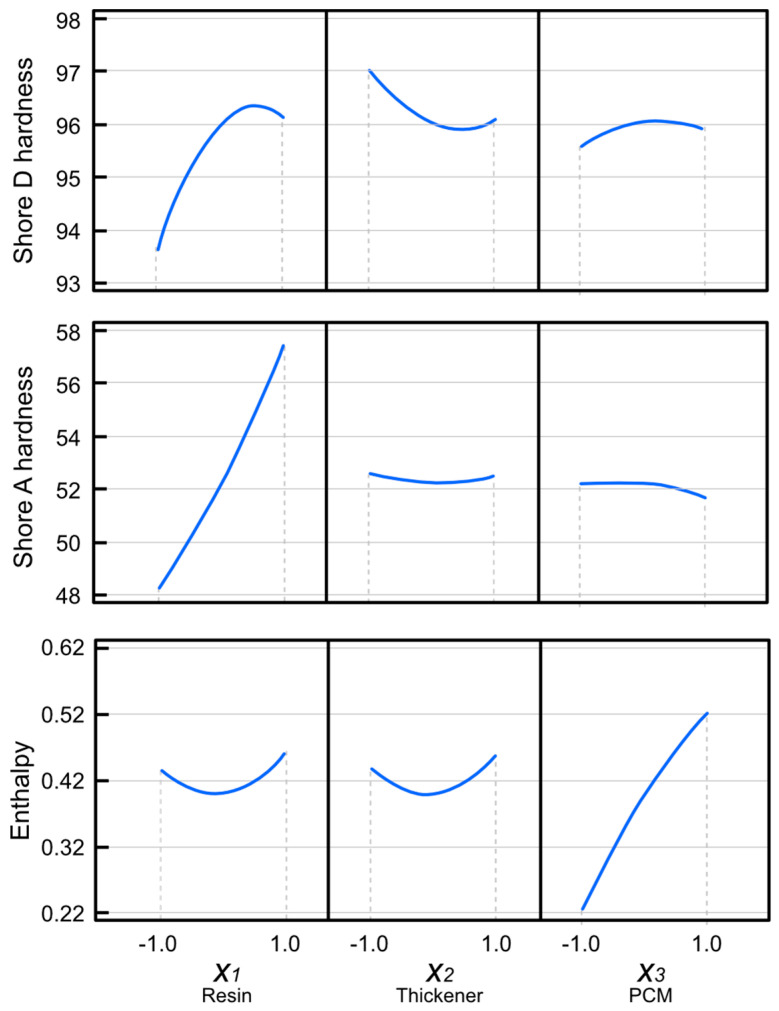
Graphs of principal effects for Shore D and Shore A hardness and enthalpy.

**Figure 7 materials-15-06967-f007:**
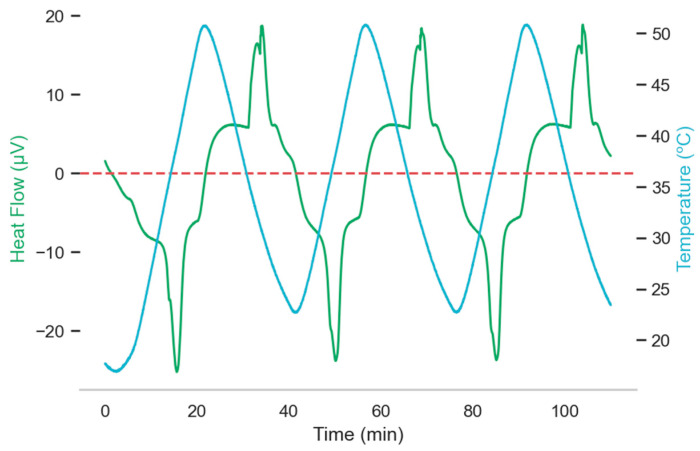
Endo- and exothermic transformations in two reversible cycles.

**Figure 8 materials-15-06967-f008:**
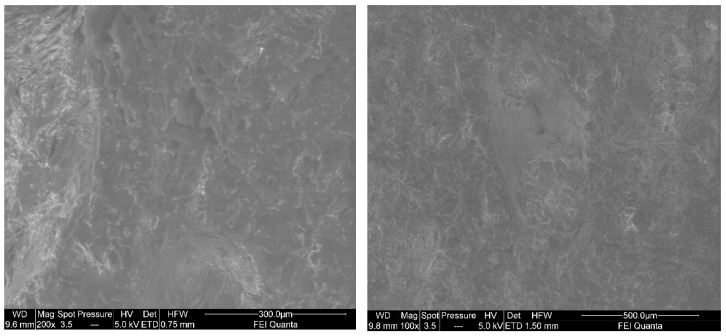
Secondary electron topographic images of the sample section.

**Table 1 materials-15-06967-t001:** Shore D results for previous insole characterization.

Sampling Point	Insole 1	Insole 2	Insole 3
1.	45	48	57
2.	55	51	67
3.	57	54	68
4.	54	55	65
5.	51	47	61
Mean value	52.4	51	63.6
CV (%)	8.9	6.9	7.2

**Table 2 materials-15-06967-t002:** Independent and dependent variables used in the study.

** *(a) Independent Variables* **			
**Variable**	**Nomenclature**	**Units**	**Range of Variation**
Resin (1050)	[R]	%	(20, 40)
Thickener	[T]	g	(0.3, 0.5)
PCM	[PCM]	g	(1, 2)
** *(b) Dimensionless, Coded Independent Variables* **			
**Variable**	**Nomenclature**	**Definition**	**Variation Range**
Dimensionless resin	x_1_	([R] – 30)/10	(–1, 1)
Dimensionless thickener	x_2_	([T] – 0.4)/0.1	(–1, 1)
Dimensionless PCM	x_3_	([PCM]–1.5)/0.5	(–1, 1)
** *(c) Dependent Variables* **			
**Variable**	**Nomenclature**	**Units**	
Shore D hardness	y_1_	Shore D	
Shore A hardness	y_2_	Shore A	
Absorbed heat	y_3_	µV/mg	

**Table 3 materials-15-06967-t003:** Operational conditions considered in this study (expressed in terms of the coded independent variables: dimensionless resin ×1; dimensionless thickener ×2; and dimensionless PCM ×3). Experimental results achieved for the dependent variables: y1 (Shore D hardness); y2 (Shore A hardness); and y3 (absorbed heat after thermogravimetry).

	Independent Variables			Dependent Variables		
Exp.	*x* _1_	*x* _2_	*x* _3_	*y* _1_	*y* _2_	*y* _3_
1	−1	−1	0	49	93.4	0.5145
2	−1	1	0	49	94.6	0.4774
3	1	−1	0	58	97.4	0.4416
4	1	1	0	57	96.4	0.6115
5	−1	0	−1	49	94	0.3126
6	−1	0	1	46.2	93.2	0.5020
7	1	0	−1	56	94.4	0.3723
8	1	0	1	59.4	97	0.5243
9	0	−1	−1	53	97.6	0.1893
10	0	−1	1	51.2	95.8	0.6708
11	0	1	−1	53	95.6	0.1970
12	0	1	1	52	94.8	0.6211
13	0	0	0	52.2	96	0.3884
14	0	0	0	52.2	96	0.438
15	0	0	0	52.2	96	0.4188

**Table 4 materials-15-06967-t004:** Ascending path for Shore D/Shore A hardness and enthalpy, showing the effect of increasing the resin content in six steps of 1% increments.

**Resin (%)**	**Thickener (g)**	**PCM (g)**	**Predicted Shore D Hardness Values**
0.0	0.0	0.0000	96.0
1.0	−2.1377	0.9338	102.392
2.0	−6.2131	2.3701	134.047
3.0	−10.4371	3.7829	194.466
4.0	−14.6997	5.1868	283.881
5.0	−18.9797	6.5861	401.386
**Resin (%)**	**Thickener (g)**	**PCM (g)**	**Predicted Shore A Hardness Values**
0.0	0.0	0.0000	52.2
1.0	−0.02299	0.07579	57.651
2.0	−0.07504	0.3413	65.271
3.0	−0.1416	0.7052	75.508
4.0	−0.2161	1.1243	88.535
5.0	−0.2953	1.5771	104.431
**Resin (%)**	**Thickener (g)**	**PCM (g)**	**Predicted Enthalpy Values**
0.0	0.0	0.0000	0.4
1.0	1.0	8.2553	−0.303
2.0	2.0	16.3357	−4.805
3.0	3.0	24.5669	−13.267
4.0	4.0	32.8108	−25.654
5.0	5.0	41.0529	−41.964

**Table 5 materials-15-06967-t005:** Optimized values for Shore D/A hardness and enthalpy, expressed as coded and uncoded values.

**Shore D Hardness**		
**Factor**	**Optimum value (coded)**	**Optimum value (uncoded)**
**Resin**	0.9999	**~40**
**Thickener**	−0.9703	**0.48**
**PCM**	1.0	**2**
**Shore A Hardness**		
**Factor**	**Optimum value (coded)**	**Optimum value (uncoded)**
**Resin**	1.0000	**40**
**Thickener**	−1.0000	**0.3**
**PCM**	0.9994	**~2**
**Enthalpy**		
**Factor**	**Optimum value (coded)**	**Optimum value (uncoded)**
**Resin**	−1.0	**20**
**Thickener**	−1.0	**0.3**
**PCM**	0.9522	**1.9**

## Data Availability

The data that support the findings of this study are available from the corresponding author, upon reasonable request.

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
