# Peer review of "Effect of Phase-Change Materials on Laboratory-Made Insoles: Analysis of Environmental Conditions"

_materials, 2022, doi:10.3390/ma15196967_

Round 1

Reviewer 1 Report

The paper may be improved after some revisions:

1.      In Fig. 5, it was found that resin is the most influencing variable for shore D hardness, with increasing hardness towards higher resin concentration. However, Fig. 6 shows the negative effect of resin on hardness, as surface shows a decreasing slope as resin concentration increases. Why the results are different?

2.      The two pictures in Fig. 8 seem different from each other. What does each one represent? Why there are the differences?

3.      The text in some figures (e.g. Fig. 5) are unclear.

Author Response

Please, read the attached file.

Reviewer 2 Report

PCM was introduced into insoles to improve its temperature-regulation ability to achieve comfort in this paper. Orthogonal experimental design was carried out to find out the effect of each component and the optimal combination. However, the purpose of this paper shall be to find a balance solution for hardness and comfort by temperature regulation, not to find the hardest formula. This paper mainly focuses on hardness and heat exchange property. But for comfort, it is not the hardest, the best. And, even without the experiments, it is well known that resin dominates the hardness and PCM dominates the heat exchange. The effect analysis is thus meaningless.  The temperature comparison in shoes with or without PCM shall be reported. This paper is rather a technical report  than an academic research paper.

Author Response

Please, read the attached file.

Reviewer 3 Report

This is an intriguing ms providing real interest in a well defined application.

IT would be very helpful to the reader  if the authors addressed the following areas

1) Provide details of the experimental methods employed - it is very limited in the submitted ms

2) I note that heat exchange is measured in units of  uV/mg which is meaningless to the reader. I assume this is the output of the device used, which could be calibrated.

3) It would be helpful to have physiological based specification for the insole - what is the aim of the work?

4) I suspect that it would be helpful to have a simple thermal model of the foot, the insole and the shoe, to understand how the heat is transferred to the environment, since the insole is an enclosed environment. I am assuming the ultimate aim is to regulate the temperature of the foot, which of course the body is doing at the same time.

5. This ms describes work which is related to paraffin loaded resins and their properties, but as the title specifically identifies the application is with insoles for shoes and I think the  reader would benefit if this second strand had a great visibility in the ms, so that the real potential can be identified.

Author Response

Please, read the attached file.

Reviewer 4 Report

 The authors presented the phase change materials for wearable thermal management, insoles. I think the study will contribute its part for the development of phase change composite for large-scale implementation. It can be considered publication after considering the following major points. Detailed below:

1) The Abstract is not worthy for reading. It needs extensive revision. Please avoid unnecessary abbreviations, it can be expressed in other terms too. Meanwhile, please present the main findings with relevant descriptions.

2) The size of the introduction is good-enough but needs up-to-date information. What is new in this work? 

3) I couldn't find the phase change enthalpy or heat storage capacity of composite PCMs. How do you confirm the phase transition?

4) Figs. 5 and 6 are not clear. 

5) As indicated, while heating the PCM exhibited a phase transition. Is the author suggested PCM without supporting addition? How stable is it?

General comment

1, Please thoroughly revise the whole manuscript. Needs critical English editing.

2. Avoid unnecessary abbreviations, upper and lower letter, symbols etc. 

3. Please make sure that once the abbreviation is defined, please use the abbreviation for the next description. For example, the term "phase change materials" is abbreviated as "PCMs" in introduction section. But the author repeated in conclusion section. The same is true for others.  

Author Response

Please, read the attached file.

Round 2

Reviewer 4 Report

The authors tried to consider my previous comments, which is appreciated. However, the manuscript still needs extensive editing including the English language. Therefore, I still prefer to give a room for the authors to revised their manuscript. Please pay much attention at least the following points:

1, The Abstract is not acceptable in the current form. Unfortunately, the current version of the revised manuscript is amended in an incorrect way. I strongly suggest the authors to refer the journals guideline and similar papers.

2, The revision in the introduction is good. But still needs up-to-date information.

3. How much is the latent heat of the insole?

4. Strongly recommend to revise the English. 

Author Response

The authors tried to consider my previous comments, which is appreciated. However, the manuscript still needs extensive editing including the English language. Therefore, I still prefer to give a room for the authors to revised their manuscript.

The new additions are highlighted in yellow and the English corrections in green.

Please pay much attention at least the following points:

Q1. The Abstract is not acceptable in the current form. Unfortunately, the current version of the revised manuscript is amended in an incorrect way. I strongly suggest the authors to refer the journals guideline and similar papers.

The Abstract has been rewritten following the best practices.

Q2. The revision in the introduction is good. But still needs up-to-date information.

Following reviewer suggestions, up-to-date information has been added to the introduction section. As a consequence, thirteen new references were added, all of them published from 2020 to 2022. The new paragraph can be checked in page 2 (lines 51-68) and we find that it helps to focus the interest and novelty of this paper. So, advances in footwear thermal regulation have been relatively unattended in the bibliography, whereas clothing has received more attention. Instead, there are references claiming for more attention to footwear, especially in situations like firefighting or orthopedic, among others, where the needs are more demanding and the references even more scarce. We believe that this paper may be of interest to improve thermal regulation in footwear and the results presented will contribute to the advance in the research in this area.

Q3. How much is the latent heat of the insole?

According to the reviewer’s suggestion, the latent heat of the insole was highlighted. The manuscript now says (lines 429 to 435): The weight of an orthopedic insole will vary, depending on foot widths and type of corrective treatment: an average weight of 50 grams of resin per insole was estimated. Sample 12, the selected sample, contained 20% by weight of PCM. Considering the properties of paraffin, that weight implies a latent heat in the orthopedic sole of approximately 48 kJ/kg. These data are an estimation and the final prototype still has to be manufactured and tested.

Q4. Strongly recommend to revise the English. 

Aware of both the limitations on time and the voluntary nature of the review process, one can be nothing other than thankful to reviewers for their efforts to improve the quality of scientific papers.

Nevertheless, whenever a Reviewer of a scientific journal comments on the quality of the English language, then reasoned opinions to support the chosen examples are always welcome.

The translator has therefore certified the most recent version of the paper, stating that in his opinion the use of the English language in the paper is of an acceptable standard for publication in a scientific journal. 
